# Nerve Reconstruction Using ActiGraft Blood Clot in Rabbit Acute Peripheral Injury Model: Preliminary Study

**DOI:** 10.3390/bioengineering11040298

**Published:** 2024-03-22

**Authors:** Shimon Rochkind, Sharon Sirota, Alon Kushnir

**Affiliations:** 1Sackler School of Medicine, Tel Aviv University, Tel Aviv 6997801, Israel; 2RedDress Ltd., Pardes Hana 3701142, Israel; sharon@reddress.co.il (S.S.); alon@reddress.co.il (A.K.)

**Keywords:** peripheral nerve injury, ActiGraft blood clot, nerve regeneration

## Abstract

This preliminary study aimed to investigate an ActiGraft blood clot implant (RedDress Ltd., Pardes-Hanna, Israel) attempting to treat and induce the regeneration of a completely injured peripheral nerve with a massive loss defect. The tibial portion of the sciatic nerve in 11 rabbits was transected, and a 25 mm nerve gap was reconnected using a collagen tube. A comparison was performed between the treatment group (eight rabbits; reconnection using a tube filled with ActiGraft blood clot) and the control group (three rabbits; gap reconnection using an empty tube). The post-operative follow-up period lasted 18 weeks and included electrophysiological and histochemical assessments. The pathological severity score was high in the tube cross sections of the control group (1.33) compared to the ActiGraft blood clot treatment group (0.63). Morphometric analysis showed a higher percentage of the positive myelin basic protein (MBP) stained area in the ActiGraft blood clot group (19.57%) versus the control group (3.67%). These differences were not statistically significant due to the small group sizes and the large intra-group variability. The results of this preliminary study suggest that the application of an ActiGraft blood clot (into the collagen tube) can enable nerve recovery. However, a future study using a larger animal group is required to achieve objective statistical results.

## 1. Introduction

Peripheral nerve injury (PNI) is a global clinical issue that negatively impacts the quality of life of affected individuals. Damage to peripheral nerves can disrupt the communication between the central nervous system (CNS) and peripheral organs, causing a loss of function in both sensory and motor nerves. Such injuries frequently lead to painful neuropathies, reducing motor and sensory capabilities and significantly interfering with daily activities. Traumatic accidents are the primary causes of peripheral nerve injuries [1,2].

### 1.1. Strategies for Treating Complete Peripheral Nerve Injuries

While peripheral nerves possess the ability to regenerate following injury, this natural repair process may not be adequate for achieving optimal functional recovery. The treatment of PNI and the likelihood of a complete return of nerve function is based on the type and severity of the underlying disease or injury [3,4]. The Sunderland or Seddon classification scheme is the most widely used method for characterizing PNI in a clinical setting [5]. Both Seddon and Sunderland classifications offer the opportunity to establish a correlation between clinical symptoms and histological alterations observed in the nerve structure at the time of injury [6,7]. Seddon’s classification comprises three levels: neuropraxia, axonotmesis, and neurotmesis. Meanwhile, the Sunderland classification system delineates five degrees of nerve damage (I–V) on the basis of histological findings. In grade I injuries, akin to neuropraxia, the axon’s integrity is preserved. Grade II injuries involve the cutting off both the axon and myelin, with the endoneurium, epineurium, and perineurium remaining undamaged, facilitating axon regeneration. Grade III injuries maintain the integrity of the epineurium and perineurium. In grade IV injuries, the endoneurium and perineurium are damaged, but the epineurium stays intact. Grade V injuries, equivalent to neurotmesis, entail the disruption or laceration of the nerve trunk. These fibrosis injuries lead to neuroma or neuropathic changes, necessitating nerve reconstruction [8,9,10,11]. Key considerations in the repair of complete peripheral nerve injuries include the length of the nerve gap, the duration between the injury and repair, and the patient’s age. The primary approach involves tension-free end-to-end repair, which is accomplished by suturing the nerve stumps together using epineural and/or group fascicular techniques. In cases where a substantial nerve gap cannot be hindered and must undergo end-to-end repair, the use of an autologous peripheral nerve graft or nerve tube becomes necessary. These serve as bridges between the nerve stumps, facilitating axonal regrowth and providing essential support.

### 1.2. Peripheral Nerve Grafts

#### 1.2.1. Autologous Nerve Grafts

The prevailing standard of care for complete peripheral nerve injuries (PNIs) encompasses the “gold standard” of autologous nerve grafts (ANGs), FDA-approved hollow tubes, and decellularized nerve allografts [12]. Nonimmunogenic nerve grafts, which are considered the gold standard for nerve damage reconstruction, are sourced from the same patient [13]. Similarly, the most commonly employed and acknowledged strategy to bridge nerve gaps is the use of autologous nerve grafts (ANGs), which are harvested from a different location within the patient’s body. These grafts offer structural guidance, facilitating axonal progression from the proximal to the distal nerve stumps. However, the utilization of ANGs presents notable drawbacks, such as the need for a second surgery site to harvest tissue, leading to donor-site morbidity and loss of function [14]. Currently, ANGs are limited to critical nerve gaps of approximately 5 cm in length [15]. The challenges of using ANGs include mismatches in donor nerve size and fascicular inconsistency between the autograft and the recipient site’s proximal and distal stumps. Additional disadvantages involve the potential for infection and the formation of painful neuromas. These limitations have prompted researchers to explore and develop alternative approaches, including creating novel nerve conduits to treat peripheral nerve injuries.

#### 1.2.2. Allografts

The utilization of nerve allografts involves employing tissues obtained from a cadaver to bridge peripheral nerve lesions. These allografts provide structural support and serve as sources of viable donor-derived Schwann cells, facilitating an axon connection at the proximal and distal ends for the re-innervation of target tissues or organs [16]. However, the application of allografts comes with inherent limitations, notably immune rejection, the risk of cross-contamination, and potential secondary infections. To address some of these challenges, nerve allografts can undergo processing through repeated freeze–thaw cycles, irradiation, and decellularization with detergents.

#### 1.2.3. Nerve Tubes

Cylindrical nerve grafts have been employed in nerve repair since as early as 1879, marked by the initial use of a bone tube as a nerve conduit [17]. Nerve guidance conduits serve as biomaterial-based scaffolds that facilitate nerve repair and regeneration, bridge nerve defects, and guide axon regrowth to the appropriate distal targets. Currently, there are 11 FDA-approved conduits for treating peripheral nerve injuries (PNIs), which are produced from biomaterials of both natural and synthetic origins. The advantages of using nerve guidance conduits over autologous nerve grafts include procedural simplicity, a substantial reduction in surgery time, and the absence of sensory loss or cosmetic defects resulting from donor site intervention. However, a notable drawback of nerve guidance tubes is their limited ability to bridge nerve losses longer than 2–2.5 cm.

#### 1.2.4. Autologous Blood Clots

Research on the clinical and scientific literature has indicated that the application of autologous blood clots can promote healing, enhance the movement of essential substrates, reduce bioburden, and stimulate angiogenesis. The topical application of autologous blood clot tissue serves as a potent cellular- and tissue-based therapy, demonstrating safety and effectiveness. While the primary component is blood, the autologous clot tissue forms a structure that functions as a biological delivery system over several days, regulating the release of cytokines and growth factors [18]. The use of topically applied blood clots has gained significant interest as they offer stromal matrices containing viable cells that are autologous, biocompatible, biological, and consistent with a metabolically active scaffold. This approach has proven to be safe, effective, and cost-efficient. Autologous whole blood clots (AWBCs) are employed at the point of care for the treatment of cutaneous wounds, demonstrating safety and efficacy in managing chronic wounds of various etiologies in patients who require wound care [19].

#### 1.2.5. Fibrin as a Filler for the Nerve Tube

Researchers, such as Galla et al. [20], have pioneered the use of fibrin as a filler matrix within conduits for nerve repair. Galla et al. utilized a fibrin matrix enriched with Schwann cells and neurotrophic factors to bridge a 10 mm gap in the facial nerves of rats. However, their method yielded outcomes inferior to autograft repair [20]. In another study, Choi et al. [21] used an autologous vein filled with a fibrin matrix to bridge 17 mm peroneal nerve defects in rabbits. Some studies in mouse models reported comparable or even superior results to autologous nerve grafts when using conduits filled with fibrin [22,23]. Schuh et al. [24] employed rolled collagen–fibrin-engineered neural tissue within a silicone tube to bridge an 8 mm defect in the sciatic nerves of rats, observing increased axon numbers in midgrafts and distal parts compared to conduits filled with collagen alone. Carriel et al. [25] added mesenchymal stem cells to a fibrin–agarose gel within a collagen nerve conduit, resulting in enhanced nerve regeneration compared to conduits filled with fibrin–agarose gels alone, although this method was inferior to autograft repair [26]. Fibrin hydrogel within a chitosan conduit, used to bridge a 10 mm gap in the sciatic nerves of rats, demonstrated efficacy comparable to autologous nerve grafts [27]. The literature review highlights that most experimental work was conducted on rats with gaps smaller than critical-size defects, prompting further investigation into the use of ActiGraft blood clots for treating critical-size defects in a rabbit peripheral nerve injury model.

## 2. Materials and Methods

### 2.1. Animals and Surgical Procedure

Animal experiments were approved by the Council for Experiments of Animal Subjects at the Israeli Ministry of Health and adhered strictly to the Animal Care guidelines. The animals were housed under standard conditions (a room temperature of 20–24 °C; a relative humidity (RH) of 30–70%; a 12:12 h light/dark cycle; and 15–30 air changes/h in the study room). Food and water were provided ad libitum.

### 2.2. Rabbit PNI Model

Sedation and Anesthesia: Eleven New Zealand white female rabbits weighing 2.5–3 kg were anesthetized with an IM injection of 15 mg of ketamine/kg + 0.25 mg of domitor/kg to the back muscles. Then, the rabbits were placed on the surgery table and connected to an anesthetic machine that delivered isoflurane inhalation via a facemask (2% in oxygen at a flow rate of 0.8–1.2 L/min). The area of the surgery was shaved, washed with ethanol and polydine solution, and then covered with a sterile sheet to ensure sterile conditions.

### 2.3. Induction and Repair of Peripheral Nerve Injury (PNI)

Experimental Design and Surgical Technique: The operation on the tibial portion of the sciatic nerve was carried out on the left hindlimb. We used a recently published rabbit model for peripheral nerve reconstruction, which prevents the autophagy that often happens in rabbits after experimental sciatic nerve injury [28]. The rabbit was put in a prone position, the hindlimbs were abducted, and the skin over the lateral and caudal aspects of the limb up to the lumbar midline was sheared. The sciatic nerve was accessed from the lateral side of the thigh. An incision of about 6 cm in length was made. The fascia was sharply divided, and the two muscles (biceps femoris and semimembranosus) were bluntly retracted to enable access to the sciatic, peroneal, and tibial nerves. Using a microscope, the tibial portion of the sciatic nerve was exposed and separated from the peroneal portion. A 1 cm nerve segment of the tibial portion was removed, and a 2.5 cm gap was created between the two ends (Figure 1).

The proximal and distal ends of the nerve were fixed into 3 cm of the NeuraGen^®^ nerve guide tube (Integra LifeSciences, Princeton, NJ, USA), creating a 2.5 cm gap between the two ends, and then micro-surgically reconnected using 9-0 epineurium sutures (Figure 2).

Then, the muscles were sutured using 3-0 vicryl threads, and the skin was closed using special metal staples. The post-operative period was 18 weeks.

The animals were divided into two groups: In the first group (three rabbits, control group), the nerve was reconnected using an empty NeuraGen tube. In the second group (eight rabbits, treatment group), the nerve was reconnected using a NeuraGen tube filled with ActiGraft blood clot (RedDress Ltd., Ponte Vedra Beach, FL, USA) (Table 1).

#### Method of Blood Sampling Used to Prepare the ActiGraft Blood Clot

Approximately 30–60 min prior to surgery, two milliliters of whole blood samples were obtained from the central ear vein/artery of each animal using an appropriately sized needle and placed into sterile Eppendorf tubes, which were filled in advance with 300 μL of acid–citrate–dextrose adenine (ACDA), to obtain one of the test item’s components (in the group that used a tube filled with ActiGraft blood clot). Following the surgical procedure, citrated blood was activated ex vivo by drawing it from the Eppendorf tubes into a procedure syringe containing 7 mg of kaolin powder and 15 mg of calcium gluconate powder (RedDress Ltd., Israel) and immediately administered through the nerve guide (NeuraGen^®^), filling it with 2 mL of whole blood clot.

### 2.4. Organ/Tissue Collection and Fixation

Samples of sciatic nerves (implant site, n = 11) from 11 rabbits were harvested, fixed in a fresh 4% PFA solution, transferred to Patho-Logica in the fixative, and kept for 48 h for further fixation. Then, the tissues were trimmed, put in embedding cassettes, and processed routinely for paraffin embedding. Three cross slices were taken, namely proximal from the lesion, med in the lesion, and distal from the lesion. In addition, two longitudinal sections were taken from the junction between the lesion and the normal tissue. For each animal, one cassette was created.

#### 2.4.1. Slide Preparation

Paraffin sections (4 microns thick) were cut, put on glass slides, and stained with hematoxylin and eosin (H&E) for general histology, and myelin basic protein (MBP) immunohistochemistry (IHC) was used as a marker for nerve regeneration.

The slides were subjected to histopathological evaluation.

#### 2.4.2. Light Microscopy Photography

Pictures were taken using an Olympus microscope (BX60, serial no. 7D04032) equipped with a camera (Olympus DP73, serial no. OH05504) at objective magnifications of ×10 and ×20.

#### 2.4.3. Histopathological Evaluation

The H&E-stained slides were examined, described, and scored by the study pathologist using a five-point semi-quantitative (SQ) grading scale to determine the severity of the histopathological changes (Schafer et al., 2018) [29], as follows:Grade 0—the tissue appears normal.Grade 1—minimal pathological findings.Grade 2—mild pathological findings.Grade 3—moderate pathological findings.Grade 4—severe pathological findings.

The histopathological evaluation included a comparison between the animals in the treatment and control groups. The pathological findings were described, scored, and demonstrated in representative histological pictures.

#### 2.4.4. IHC of MBP for Myelin Staining (Per a ×20 Field)

A semi-quantitative analysis of the intensity of the immunohistochemical staining reaction for myelin was performed using the following scoring scale:Grade 0 = no positive staining reaction.Grade 1 = only a few cells are immune positive (<5 cells).Grade 2 = very mild immune reaction (5–15 cells).Grade 3 = mild immune reaction (15–25 cells).Grade 4 = moderate immune reaction (25–50 cells).Grade 5 = significant immune reaction (>50 cells).

e. Digital morphometry (×10 mag)

A morphometric analysis of % MBP positive staining (out of total area) was performed via brightness and color-based segmentation using the MATLAB software. The borders of the nerve were determined manually.

#### 2.4.5. Electromyography (EMG) Examinations

Both hind legs were examined to demonstrate the difference/s between a “naïve” leg and an operated one. The parameters examined were as follows:Amplitude = the peak of the action potential of the electrical signal, measured in microvolts (μV);Latency = the time for the action potential to take place, measured in milliseconds (ms).

#### 2.4.6. Statistics

The non-parametric semi-quantitative scores were subjected to the Mann–Whitney U test. The results of the digital morphometric analysis were subjected to a non-paired Student’s *t*-test. The significance level was *p* = 0.05 for both tests.

## 3. Results

The rabbits received treatment according to the details outlined in Table 1 and underwent an 18-week follow-up period. Electrophysiology assessments were conducted during this period. 

### 3.1. Mortality Incidents

No mortality incidents were recorded in any of the animals during the entire observation period, except for one non-treatment-related mortality that occurred during the surgical procedure.

### 3.2. Electromyography (EMG) Examinations

The EMG examinations of the gastrocnemius muscle showed normal values of amplitude and a normal latency period at baseline. At the 2-week post-op time point, the examination showed that the expected damage to the area was determined and represented by low values of amplitude and a long latency period. At the third examination timepoint, prior to termination, the amplitude values were still low, and a long latency period was noted as well; therefore, no improvement could be demonstrated in terms of the EMG values.

### 3.3. Histopathology

Eighteen weeks after the surgery, the tibial segment of the sciatic nerve was extracted and divided into three sections: proximal to the injury site, the middle portion (including the injury area), and distal to the injury. These tissues were then preserved in 10% formalin, subjected to processing, and embedded in paraffin blocks.

#### 3.3.1. H&E Staining

After collecting the tibial segment of the sciatic nerve for an immunohistochemistry analysis in the weeks following the treatment, we evaluated the sample quality by conducting H&E staining. Overall, the sections stained with H&E exhibited minimal lymphocytic infiltration in the experimental cross sections. A moderate infiltration composed of macrophages and a few multinucleated giant cells was only observed in one rat from the control group (treated with an empty tube). Furthermore, no other pathological changes were found (i.e., necrosis and vascular complications). The pathological severity scores (H&E) were high in the medial (area of the tube) cross sections of the control group (1.33) compared to those of the treatment group (0.63). Mild changes were noted in the proximal and distal cross sections, with a slight difference between the groups (Figure 3). 

#### 3.3.2. Myelin-Based Protein (MBP) Staining

Subsequently, we utilized myelin-based protein (MBP) staining to assess nerve recovery in the samples. The myelin staining demonstrated high intensity in the proximal cross-section, revealing densely positive myelin fibers comparable to the structure of a normal nerve.

In the middle cross section (area of the tube), an improvement in the treatment group compared to the control group was noticed, as the morphology of the repaired nerve fibers appeared loose, with a diameter of 4–6 microns filling the entire space within the peripheral membrane that surrounded the treated area. The scores of the MBP staining in the distal parts of the nerves ranged from mild to moderate. The MBP-stained middle cross sections (area of the tube) showed clear increases in the staining scores in the treatment group (2.71) compared to the control group (1.33). The scores of the proximal and distal cross-sections of the treatment and control groups were almost the same (4.5 versus 4 for the proximal part and 2.14 versus 2 for the distal part).

The morphometric analysis indicated that the most prominent differences between the groups were in the middle sections (area of the tube), with a higher percentage of positive MBP stained area in the treatment group (19.57%) versus (3.67%) in the control group. 

As anticipated, the proximal sections exhibited a higher percentage of the positive stained area, particularly in the treatment group (47.75%) compared to the control group (35.67%). Conversely, in the distal sections, the positive stained area was greater in the control group (22%) than in the treatment group (13.14%). However, due to the considerably high intra-group variances, none of the differences between the groups reached statistical significance (Figure 4 and Figure 5).

## 4. Discussion

Peripheral nerve injuries can lead to profound and enduring functional impairments. Incidents such as car accidents, gunshot wounds, deep cuts, oncological resections, and other traumas often result in substantial damage to peripheral nerves, causing a loss of function in limbs with limited prospects for recovery. In cases where larger nerve gaps (20 mm or longer in humans) exist, the prevailing clinical gold standard for peripheral nerve reconstruction is the use of an autologous sensitive nerve graft (autograft). However, re-innervation with cutaneous sensitive nerves may not always yield satisfactory outcomes, as it is necessary to include motor fibers in the bridging nerve grafts [30]. While nerve tubes present certain advantages over autologous nerve grafts, their limitation in bridging gaps exceeding 2–3 cm of nerve loss restricts their widespread application in clinical practice for peripheral nerve reconstruction. Therefore, the imperative for the advancement of effective and efficient repair techniques for nerve injuries remains paramount, holding significant implications for the well-being of both civilian and military populations. Many endeavors have been undertaken to augment nerve regeneration, employing diverse tube luminal scaffolds encompassing a spectrum from collagen and laminin hydrogels to synthetic materials and collagen filaments and channels [31,32,33,34,35,36]. Despite the array of innovative modifications introduced into this landscape, the outcomes thus far have fallen short of achieving results comparable to, let alone surpassing, the efficacy of autografts. The quest for a superior nerve regeneration solution continues to be unabated, fueled by the recognition that current modifications, as evidenced by the existing literature [34,37,38], only offer limited advantages over the established autograft technique. As the scientific community endeavors to unravel the complexities of nerve regeneration, further exploration, and refinement of these tube luminal scaffold approaches are essential. This persistent pursuit of breakthroughs aims to unlock the full potential of these alternative techniques, ultimately providing a more comprehensive and effective repertoire of options to address the diverse challenges posed by nerve injuries in both civilian and military contexts. In the dynamic landscape of regenerative medicine, the ongoing quest for enhanced nerve repair techniques underscores the critical importance of advancing scientific knowledge and technological innovation to meet the pressing needs of patients and professionals alike.

A previous investigation by one of the authors into increasing nerve regeneration through a long-distance gap commenced in 2004 with the development of a composite neurotube [39]. Subsequently, a guiding regenerative gel (GRG) [40] and an antigliotic guiding regenerative gel (AGRG) matrix [41] were created. These innovations serve as vehicles for axonal growth and survival, enabling the reconstruction of peripheral nerves with substantial loss defects. Using an AGRG matrix filled in a commercial collagen tube to reconstruct a critical-size defect in a chronic model of a peripheral nerve injury (a model that simulates delayed injury in humans) leads to a notable nerve improvement at the same level as autologous nerve transplantation (gold standard) [41].

A supportive laser phototherapy treatment is currently under development with the specific goal of enhancing the efficacy of nerve reconstruction procedures. Laser phototherapy harnesses visible and near-infrared electromagnetic energy, triggering photochemical shifts for therapeutic benefits and enhanced clinical recovery outcomes [42]. It is also noted for boosting neuronal survival, fostering more regenerating axons, hastening regrowth, and promoting re-innervation of nerve endpoints, resulting in improved functional outcomes [43,44,45,46]. In a study by Rochkind et al. [47], the impact of 780 nm laser irradiation on peripheral nerve recovery post-PGA neurotube reconstruction was explored. Notably, a favorable somatosensory response was observed in 70% of irradiated rats, in contrast to a mere 30% in the control group. Furthermore, the irradiated cohort exhibited heightened sciatic functional index (SFI) scores and an increased count of myelinated axons. Separately, Shen et al. [48] utilized genipin-crosslinked gelatine neurotube reconstruction coupled with 660 nm laser treatment to address 10 and 15 mm sciatic nerve gaps in rats. Examination of the grafts’ medial segments revealed enhanced neuronal regeneration, encompassing the number of augmented nerve fibers, axon diameter, myelinated sheath thickness, and overall nerve area.

Publications over the past decade have documented the therapeutic properties of a blood clot scaffold [18,49,50]. Blood, as a carrier, contains various components such as red blood cells, white blood cells, platelets, proteins, clotting factors, minerals, electrolytes, and dissolved gases. Autologous blood clot tissue functions as a carrier and establishes a protective environment that encourages the body to employ its natural healing mechanisms in an organized manner. The scaffold formed via autologous blood clot tissue serves as a medium for the body to transform a nonhealing chronic wound into a healing acute condition. This temporary scaffold supports cell ingrowth, allowing cells within the matrix to sense both soluble factors and their physical surroundings. This well-coordinated mechanism involves signals from soluble molecules, the substrate or matrix to which the cell adheres, the mechanical forces acting on it, and interactions with other cells. By creating a scaffold resembling the extracellular matrix (ECM), an autologous blood clot facilitates the re-establishment of communication between cells, incorporates itself into the soft tissue deficit, and forms a protective barrier [18,49]. The literature review showed that fibrin has a therapeutic property as a filler matrix within a tube during peripheral nerve reconstruction [20,21,22,23,24,25,26,27]. These scientific publications suggest that using a blood clot as a filler matrix within a tube can improve nerve recovery. However, most experimental works were performed on rats with gaps smaller than critical-size defects. The published data motivated us to explore the use of an ActiGraft blood clot to address critical-size defects in a rabbit peripheral nerve injury model. The objective of this investigation was to extend the nerve gap to a critical-size defect, which we can address using an ActiGraft blood clot as a scaffold to promote axonal sprouting through the tube. The nerve gap of critical size in humans reaches approximately 3 cm. Therefore, there is an increasing inclination to explore and scrutinize larger animal models for clinical applications, particularly when assessing gaps exceeding 1.5 cm [51,52,53].

Among these models, the rabbit model is extensively employed. Our current preliminary study results suggest that using a commercial nerve conduit filled with ActiGraft blood clot tissue has the potential to support axonal growth through the critical-size defect gap. Eighteen weeks after the surgery, histopathological and histochemical observations showed signs of regeneration after using an ActiGraft blood clot scaffold compared with the empty tube (Figure 3, Figure 4 and Figure 5). Eleven nerves (proximal, middle, and distal sections) were stained with H&E to evaluate the histopathological changes and underwent MBP immunostaining in order to perform a nerve regeneration evaluation 18 weeks after surgery. Histopathological changes were observed mainly in the middle section (tube area). Nerve regeneration was higher in the ActiGraft blood tissue group than in the control group, though the differences were not significant due to the small group sizes and the large intra-group variability. 

The findings from this initial study indicate that utilizing an ActiGraft blood clot (filled into a collagen tube) may promote nerve recovery. The limitation of this study is the small number of cases on which the findings are based. Therefore, a more extensive study involving larger animal groups is necessary to attain statistically conclusive results.

## 5. Conclusions

Addressing lengthy nerve gaps poses a significant challenge marked by slow regeneration rates and often incomplete recovery. To meet this challenge, an ongoing exploration of innovative concepts is essential to address longer nerve deficits. In our preliminary study, an ActiGraft blood clot within a collagen tube shows promise in facilitating nerve restoration. However, substantiating these promising indications requires a more extensive study involving a larger and more diverse animal cohort. Conducting such research is crucial to achieve conclusive statistical results and advance our understanding of ActiGrafts’ efficacy in overcoming the complexities associated with repairing extended nerve gaps. This rigorous scientific inquiry contributes additional valuable insights to the ongoing pursuit of innovative solutions for enhancing nerve regeneration and recovery.

## Figures and Tables

**Figure 1 bioengineering-11-00298-f001:**
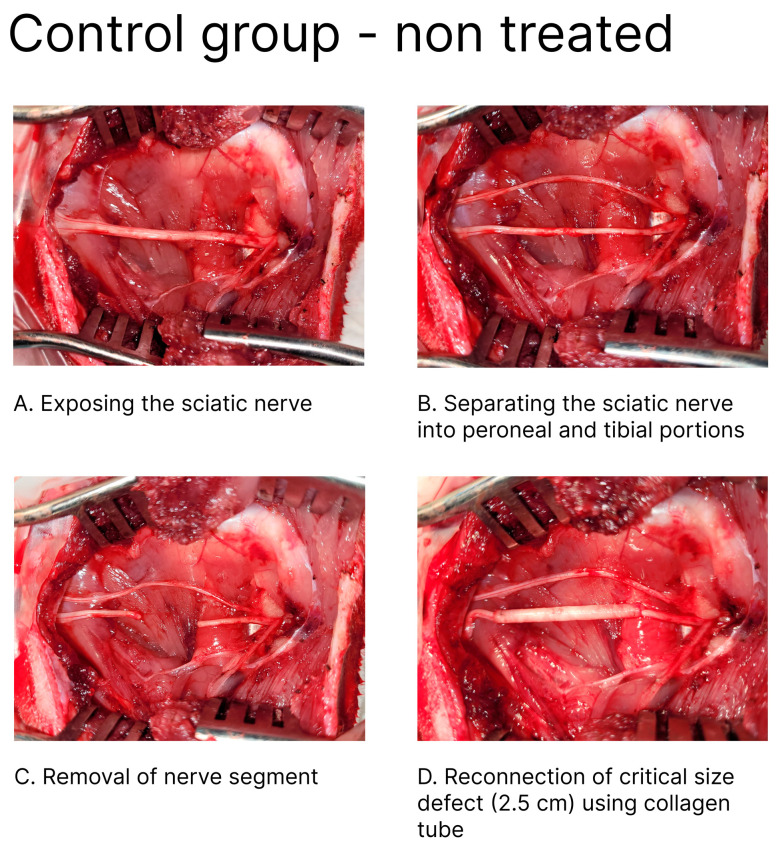
A peripheral nerve injury and reconstruction model. After exposing the sciatic nerve (**A**), the nerve is separated into two portions: peroneal and tibial (**B**). Then, the tibial portion segment of the nerve is removed, creating a 25 mm gap (**C**), followed by reconnection using an empty collagen tube (**D**).

**Figure 2 bioengineering-11-00298-f002:**
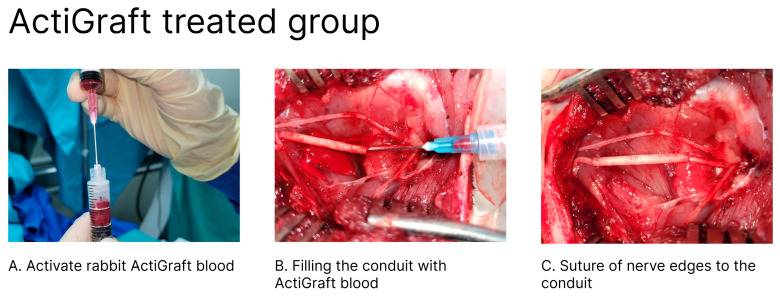
Treatment procedure: After the ActiGraft is prepared (**A**), the ActiGraft is injected into the empty tube using a syringe (**B**). Then, the conduit is sutured to the nerve ends (**C**).

**Figure 3 bioengineering-11-00298-f003:**
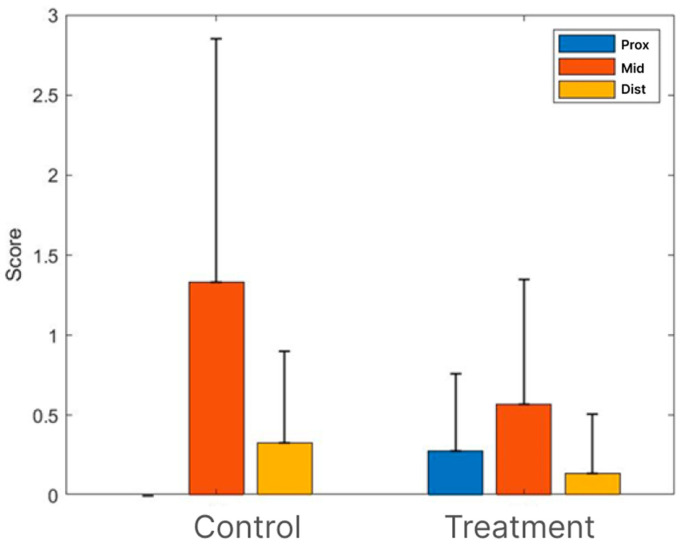
Sciatic nerve (H&E). Histopathological severity scoring in different groups (mean + SD).

**Figure 4 bioengineering-11-00298-f004:**
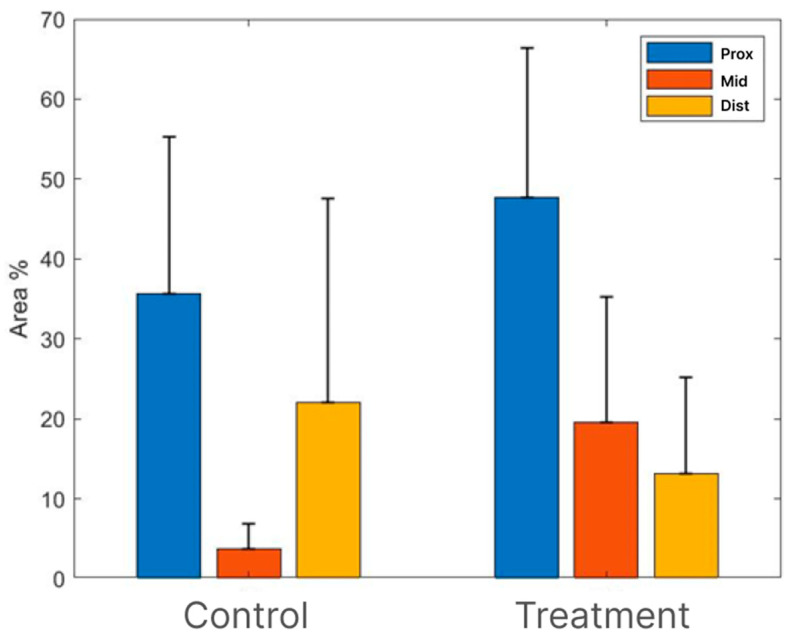
Sciatic nerve (MBP). Morphometric analysis of positive stained area (%) in different groups (mean + SD).

**Figure 5 bioengineering-11-00298-f005:**
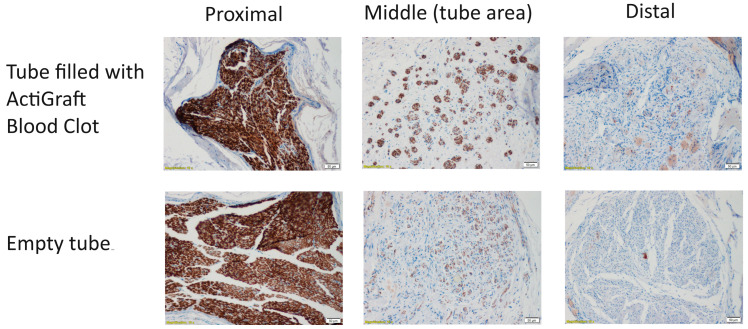
Representative histopathological photographs of MBP staining. The proximal, middle, and distal cross sections of the tibial portion of the sciatic nerve treated with ActiGraft blood clots and with empty tubes are displayed (magnification ×10).

**Table 1 bioengineering-11-00298-t001:** Rabbit acute PNI experimental design.

Treatment	N
1. NeuraGen^®^ Nerve Guide	3
2. NeuraGen^®^ Nerve Guide filled with ActiGraft Blood Clot	8

## Data Availability

The data are contained within the article.

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
