# Peer review of "Nerve Reconstruction Using ActiGraft Blood Clot in Rabbit Acute Peripheral Injury Model: Preliminary Study"

_bioengineering, 2024, doi:10.3390/bioengineering11040298_

Round 1
Reviewer 1 Report
Comments and Suggestions for Authors
The scientific paper "Nerve Reconstruction Using ActiGraft Blood Clot in Rabbit Acute Peripheral Injury Model – Preliminary Study" aimed to investigate an ActiGraft Blood Clot implant attempting to treat and induce regeneration of completely injured peripheral nerve with massive loss defect.
It can be considered that:
1) Remove the titles from the abstract. Please note in the instructions to authors: The abstract should be a single paragraph and follow the style of structured abstracts, but without headings;
2) Adjust line 36 (space);
3) In the introduction, 1.1, describe the types and classification of nerve injuries;
4) Reformulate 1.2.5. It is simple to describe previous articles. It should be described in the authors' own language, citing the numbers of the articles used for references;
5) In the methodology, the materials used must include the manufacturer, city and country;
6) Figure 1 is poorly defined and should be at least 300 dpi and increased in size for better visualization;
7) In figure 1, replace 1,2,3,4 with A,B,C,D;
8) Consider the same suggestions for figure 2 (in relation to figure 1);
9) Was the histopathological evaluation "blind"? A single evaluator?
10) In 2.4.5, is it correct: The significance level was p=0.05 for both tests? Or p ≤ 0.05?
11) 3.2. Electromyography (EMG) Examinations should be described in the methodology and its numerical results demonstrated in 3.2;
12) In figure 3, inserting letters or symbols showed statistically significant differences between the evaluated groups;
13) Remove italic from line 272;
14) In figure 4, inserting letters or symbols showed statistically significant differences between the evaluated groups;
15) Increase the size of the images in figure 5 and insert scale bar;
16) Adjust line 354;
17) In discussion, mention the beneficial effects of low-power laser photobiomodulation in peripheral nerve regeneration (example: 10.3390/bioengineering5020044)
18) Insert the limitations of the study at the end of the discussion.
19) The number of references is low to scientifically base the study. I suggest a minimum of 50 references. Increase the discussion.
Comments on the Quality of English LanguageModerate editing
Author Response
Dear Reviewer,
Thank you for taking the time to read and review our manuscript.
We value your input and adjusted the manuscript according to your comments.
Sincerely

Reviewer 2 Report
Comments and Suggestions for Authors
The submitted manuscript investigates the use of ActiGraft blood clot in rabbits to treat and induce regeneration of completely injured peripheral nerve with massive loss defect. The article is concise and clear. The article has the following comments:
1. Clearly show the scale bars for the images in Figure 1, Figure 2 and Figure 5.
2. Explain the large error bars in Figure 3 and Figure 4.
3. Briefly describe the results of electromyography (EMG) examinations during nerve regeneration.
Comments on the Quality of English LanguageThe article needs minor revision for language and grammar.
Author Response

(The authors gave the same response as above.)

Round 2
Reviewer 1 Report
Comments and Suggestions for Authors
No comments. Thanks.
Comments on the Quality of English LanguageMinor editing